# Investigation of the Seasonal Variation in Fat Patterning of Ellisras Rural Children and Adolescents, in the Limpopo Province, South Africa

**DOI:** 10.3390/children10071113

**Published:** 2023-06-27

**Authors:** Lusani Mulaudzi, Thandiwe N. Mkhatshwa, Mankopodi M. Makhubedu, Moloko Matshipi, Sogolo L. Lebelo, Kotsedi D. Monyeki

**Affiliations:** 1Department of Physiology and Environmental Health, University of Limpopo, Sovenga 0727, South Africa; 201709714@keyaka.ul.ac.za (L.M.); thandiwemngunomhle@gmail.com (T.N.M.); mankopodi.makhubedu@ul.ac.za (M.M.M.); moloko.matshipi@ul.ac.za (M.M.); 2School of Agriculture and Life Sciences, University of South Africa, Roodepoort 1710, South Africa; lebelol@unisa.ac.za

**Keywords:** fat patterning, seasonal variation, rural children and adolescents, South Africa

## Abstract

An increase in childhood obesity has become a global concern since childhood obesity often leads to adulthood obesity. This study aimed to investigate whether there is seasonal variation in fat patterning, and whether fat patterning is associated with seasons among the Ellisras population aged 5–15 years. A total of 1986 children and adolescents (1034 boys and 952 girls) aged 5–15 enrolled in this cross-sectional study. Skinfold measurements were obtained using standard procedures. Obesity prevalence was determined using frequency analysis. The correlation between obesity and two seasons was tested using multinomial regression analysis. The results showed that girls aged 11–15 years had significantly (*p* ≤ 0.03) larger median fat variables (triceps, biceps, and suprailiac) in spring compared to autumn. The prevalence of obesity ranged from 3–11% with boys being more obese (4–11%) than girls (3–7%) aged 5–15. Regression analysis showed a negative significant (*p* ≤ 0.001) correlation between autumn and obesity by the subscapular/(subscapular+triceps) (SST) ratio, both unadjusted −2.991 (95%CI −3.233:−2.803) and adjusted −2.897 (95% CI −4.331:−1.629). The findings of the study showed that there was seasonal variation in fat patterning among boys and girls in the Ellisras population and that fat patterning is associated with seasons.

## 1. Introduction

Fat patterning refers to the distribution of adipose tissue throughout the human body. Developed and developing countries are experiencing an increase in childhood obesity, which has emerged as a risk factor for non-communicable diseases [1]. Therefore, the World Health Organisation (WHO) emphasises the need to monitor obesity in different populations [2]. Globally, the prevalence of obese individuals among adolescents has significantly increased in the last 20 years [3]. The prevalence of obesity among African children was reported to be 8.5% and was estimated to be 12.7% in 2020 [4].

Childhood and adolescent obesity are strongly influenced by risk factors such as nutrition, physical activity, the environment, socio-environmental factors, biological factors, and socio-economic status [5]. Nutritional imbalance in early life and a fat-accumulation-related lifestyle are crucial in the development of obesity [6]. Skinfold measurement is frequently used to measure body fat in children. Skinfold ratios indicate the subjects with the highest risk of obesity [7]. Childhood obesity that continues into maturity increases the possibility of developing weight-related illnesses including hypertension later in life [7]. This makes tracking and controlling obesity in children a priority [8]. Perinatal life, infancy, early childhood, and adolescence have been reported as critical stages for the development of obesity [9].

Most rural communities in developing nations have limited financial means for food availability and rely on their agricultural practices for production. Rainfall is essential for a productive harvest, and agricultural activities are heavily reliant on it and other climatic factors, which have an impact on food production and availability. Changes in rainfall and climatic conditions then result in seasonal food shortages, whereby food production is not enough to cover the population [10]. Therefore, people change their diets and eating habits to adapt to food shortages because of seasonal variations [10]. Other than environmental factors, socioeconomic status may also affect food patterns and food intake. It is a fact that studies that analyse dietary intake in most instances ignore the possibility of seasonal variability.

There is plenty of information that suggests that seasons could affect health conditions and therefore should be strongly considered in the incidence of nutrient deficiency and various food profiles [11]. Seasonal variations that occur throughout the year can alter the body’s composition and fat distribution patterns. A study of seasonal variation in fat patterning amongst adolescents in urban areas showed that adolescents gain more weight during summer compared to winter [12]. However, there is limited information regarding weight changes among different seasons in South African children. Therefore, this study aimed to investigate seasonal variation in fat patterning in the rural Ellisras population and to investigate whether there is an association between fat patterning and seasons.

## 2. Materials and Methods

### 2.1. The Sampling Procedure

This study took place in Ellisras (now known as Lephalale), Limpopo province, South Africa. Ellisras is about 70 km from the Botswana border [13]. Most residents in Ellisras rely mostly on income from the Iscor coal mine and the Medupi power station. Fewer people are educated in this area; therefore, some depend on farming and cattle-rearing as a source of income. The spring season in Lephalale is said to be hot and reasonably dry with a daytime maximum temperature being around the average of 31 degrees Celsius (°C) while the nighttime maximum temperature is around an average of 18 °C. The autumn season is said to be mild and very dry with a maximum average daytime temperature of 26 °C and a maximum average temperature of 9 °C [14]. However, due to climate change, these weather conditions may not give a true reflection of the weather conditions in Lephalale in the year 2000 when data collection for this study took place.

### 2.2. Data Collection

This study is a subset of the Ellisras Longitudinal Study (ELS) and that is where data from this study was sourced [15,16]. A total of 1986 participants (1034 boys and 952 girls) aged 5–15 enrolled in this cross-sectional study. Measurements were taken in two seasons autumn and spring (May 2000, November 2000). The Ellisras Longitudinal Study (ELS) initially followed a cluster sampling procedure documented elsewhere [15,16]. Briefly, the study commenced with a total of 22 schools (10 preschools and 12 primary schools) that were randomly selected from 68 schools in the Ellisras area. A total of 2225 children (550 preschool learners and 1675 primary school learners) enrolled in the ELS at baseline in 1996 [15,16]. The ELS is an ongoing study and the same population that enrolled in the ELS at baseline is followed up to date.

This study used data collected in 2000 which is one of the oldest datasets from the ELS. Since the ELS follows the same participants over time, the 2000 data will allow tracking of the ELS participants from childhood into adulthood. The participants in the year 2000 were between 5–15 years of age, and currently all ELS participants are in their adulthood. Furthermore, the 2000 data did not have a lot of missing values for all skinfolds (triceps, biceps, subscapular, and suprailiac) measures in most of the participants. Participants with missing values for skinfold measurements were excluded from the analysis, with children less than 5 years and participants above 15 years excluded as well. Participants who did not submit a signed consent form were excluded also. All participants who were present on the day of data collection formed part of the study.

#### 2.2.1. Anthropometry

The International Society for the Advancement of Kinanthropometry (ISAK) standard procedure was followed for training and measurements [17]. All children and adolescents underwent skinfold measurements using a Slim Guide skinfold calliper (Harpenden, England). Skinfolds (triceps, biceps, subscapular, and suprailiac) were measured three times, and the results were rounded off to the nearest 0.1 mm. For all skinfold measures, the intra-tester and inter-tester technical error of measurements ranged from 0.2–6 mm (0.4–6.8%), respectively, to ensure the study’s validity and accuracy during data collection.

##### Triceps

Triceps were measured over the triceps muscle, on the posterior mid-acromiale-radiale line on a relaxed arm, with the shoulder joint slightly rotated externally as shown in Figure 1 [17].

##### Biceps

Biceps were measured over the triceps muscle, on the anterior mid-acromiale-radiale line of a relaxed arm, with the shoulder joint slightly rotated externally as shown in Figure 2 [17].

##### Subscapular

The subscapular skinfold was measured by first determining the under- most tip of the scapula, by palpating the interior-most angle of the scapula. The skinfold was then raised and measured at a line running laterally and down the subscapular landmark as shown in Figure 3 [17].

##### Suprailiac

The skinfold was measured on the ilio-axilla line superior to the iliocristale, with the arm abducted horizontally across the chest as shown in Figure 4 [17].

### 2.3. Data Analysis

Data analysis was computed using IBM Statistical Package for the Social Sciences (Version 29) with a significance level set at *p* ≤ 0.05. By comparing the ratio of trunk to limb body fat, central body fat distribution was identified. This was estimated utilising three equations [9].
$$ST\;ratio=\frac{Subscapular}{triceps}$$
$$SST\;ratio=\frac{subscapular}{subscupular+triceps}$$
$$SSTB\;ratio=\frac{subscapular+suprailiac}{biceps+triceps}$$

All subjects were classified as obese or normal using the 95th percentile cut-off points for the sum of four skinfolds (S4SK) for age and gender. The prevalence of obesity was determined using frequency analysis, and a nonparametric t-test was used for gender comparison. Participants were divided into age groups taking into consideration the critical stages for the development of obesity [18]. The association between obesity and seasons was investigated using the multinomial logistic regression analysis.

### 2.4. Ethical Clearance

The Turfloop Research Ethics Committee of the University of Limpopo granted ethical approval (clearance number: TREC/P/356/2017: PG) before the study was undertaken. Authorities and the principal of the school granted permission. The parents of the participants read and signed consent forms allowing participants to be part of the project.

## 3. Results

### 3.1. Characteristics of the Population

The results are shown as median and interquartile because the data were not normally distributed. Table 1 shows descriptive statistics of Ellisras rural children and adolescents measured in autumn and spring. Girls had significantly higher biceps measurements (6.75) than boys (4.0) at age 11–15 years (*p* ≤ 0.03) in spring. Girls also had a significantly higher triceps measurement (9.0) than boys (6.75) at age 11–15 years (*p* ≤ 0.05) in autumn. Girls showed a significantly higher suprailiac measurement (6.75) than boys (4.25) at age 11–15 years (*p* ≤ 0.03) in spring.

### 3.2. The Prevalence of Obesity Based on ST, SST, and SSTB Ratio

Table 2 displays the prevalence of obesity among Ellisras rural children and adolescents measured in autumn and spring. The prevalence of obesity ranged from 3–11% with boys being more obese (4–11%) than girls (3–7%) aged 5–15. The prevalence of obesity was mostly significantly higher in spring compared to autumn. Obesity by ST ratio among boys aged 11–15 years in 2000 was significantly (*p* ≤ 0.001) higher in autumn (11%) compared to spring (8%). The prevalence of obesity by SSTB among girls 5–10 was significantly (*p* ≤ 0.001) higher in spring (7%) compared to autumn (3%).

### 3.3. The Correlation of ST, SST, and SSTB Ratio with Season

Table 3 shows the multinomial regression in autumn (2000) and spring (2000). The regression analysis showed a significant negative association between seasons and obesity by ST, SST, and SSTB ratio ranging from −3.088 to −1.084. Obesity by ST ratio showed a significant (*p* ≤ 0.001) negative correlation with spring when unadjusted −2.559 (95% CI −2.738:−2.382). There was a significant (*p* ≤ 0.001) negative correlation between obesity by SST ratio and autumn both unadjusted −2.991 (95% CI −3.233:−2.803) and adjusted for age and gender −2.897 (95% CI −4.331:−1.629). Obesity by SSTB ratio showed a negative significant (*p* ≤ 0.001) correlation with spring both unadjusted −2.860 (95%CI −3.087:−2.672) and adjusted for age and gender −2.506 (95% CI −3.885:−1.346).

## 4. Discussion

This study aimed to investigate seasonal variation in fat patterning in the rural Ellisras population and to investigate whether there is an association between fat patterning and seasons. There was a significant (*p* ≤ 0.03) variation in median fat pattern variables (triceps, biceps, and suprailiac) when comparing by gender in the spring and autumn of 2000. A higher median was observed in girls compared to boys aged 11–15 years. A study reported that girls tend to reach puberty faster than boys [19]. Furthermore, girls are said to gain more fat mass than boys during puberty [20], with faster growth in height, upper arm circumference, and body weight [21]. Therefore, the higher median fat pattern observed among girls in this study may be due to the early onset of puberty among girls. The increase in median fat pattern variables in subscapular and suprailiac values indicate the accumulation of body fat [22]. The increase in median subscapular can be attributed to higher triglycerides and can result in lifestyle-related diseases such as cardiovascular diseases, which are the leading cause of mortality worldwide [23].

There was an insignificant variation in subscapular, triceps, and suprailiac values between spring and autumn among boys aged 5–10 years old. The results differ from those of the previous study which demonstrated a substantial rise in the subscapular skinfold in boys at ages 7, 9, and 10, as well as the suprailiac skinfold at age 9 [22]. Skinfold measurements provide information on changes in body fat distribution and are acknowledged as reliable measures of total body fatness [24], the technique is susceptible to random and systematic errors [25]. Therefore, this explains the difference found in this study and other studies. At all measurement sites, girls exhibit greater median skinfold variables than boys. These results concurred with the findings of the previous studies [26,27]. According to the research finding of the previous study [27], Ellisras girls had considerably larger triceps, biceps, suprailiac, and subscapular skinfold development than boys which is similar to the results from this study. 

The seasonal and daily rhythmicity of food intake, metabolism, and adipose tissue function is frequently disrupted in obese people [28]. The prevalence of obesity by ST, SST, and SSTB was mostly significantly higher in spring compared to autumn for both girls and boys, except for boys 11–15 years with respect to the ST ratio. A potential explanation for the higher prevalence in spring might be a decrease in physical activity [29]. The weather has an impact on physical activity [29]. A change in activity patterns from the exterior to the interior entertainment has been linked to an increase in childhood and adolescent obesity [30]. It was reported that in winter there is generally low physical activity and total energy expenditure [28]. The prevalence of obesity ranged from 3–11% with boys being more obese (4–11%) than girls (3–7%) aged 5–15. This contrasted with the observation of the prevalence of overweight among adolescent girls (ages 10–19 years) of the Caboclo of the Amazon basin [31]. The sex steroid hormones, which might be characterised as android patterns for men and gynoid patterns for women, could also have an impact on fat patterning among the Ellisras population. The increase in obesity among boys may be due to variations in sex hormones. A study reported that plasma hormones such as leptin and melatonin influence the development and regulation of obesity. In males, leptin decreases with the increase in sex hormones [28]. In pubertal boys, testosterone is linked to a rise in visceral fat [27]. One study from the same population (Ellisras) showed that obesity prevalence increases with age among children and adolescents [27], with studies from India, Iraq, Mexico, and the United States showing similar results [32,33,34,35]. However, this contradicts the results of this study.

The following recommendations may be utilised to interpret correlations: Low = 0.3, moderate = 0.3–0.6, and >0.6 high [36]. These recommendations may suggest a high negative correlation for ST, SST, and SSTB ratios in both autumn and spring. There was a significant negative correlation (*p* < 0.001) in ST, SST, and SSTB ratios in autumn and spring. When two variables are negatively correlated, it signifies that when one increases, the other decreases [37]. The inverse correlation indicated that as seasons change to autumn and spring, the development of obesity decreases. The correlation observed in autumn in this study contradicts those reported by an earlier study which reported an increase in obesity in autumn [38]. The negative correlation observed in spring might be attributed to the fact that spring is the time of the year when physical activity is at its peak [39]. Furthermore, it was reported that during spring there are many outdoor activities such as picnics and physical activity is very high [28].

Limitations of the study include the fact that measurements of plasma insulin, cortisol, melatonin, and leptin could have elucidated some information on the variations of the prevalence of obesity in both girls and boys. Furthermore, the study did not include an analysis of nutritional status, diet, physical activity, and sleep quality, since data collection of these variables was not yet introduced in the ELS. Future studies may investigate seasonal variation in these variables as well. Moreover, the study only investigated autumn and spring in the same year (2000) out of the four seasons known. Therefore, it is not known whether time or seasonality or both played a role in the prevalence of obesity and the correlation between obesity and seasons. Future studies may investigate this using longitudinal data and all four seasons. Although skinfold measurements have been reported to be reliable measures for obesity, the method has limitations in severely obese participants. Therefore, future studies may investigate seasonal variation using other obesity indices.

## 5. Conclusions

There was seasonal variation among boys and girls in the Ellisras rural population. A higher obesity prevalence was observed in boys rather than girls. Furthermore, a negative correlation between seasons (spring and autumn) and obesity by ST, SST, and SSTB ratio was observed.

## Figures and Tables

**Figure 1 children-10-01113-f001:**
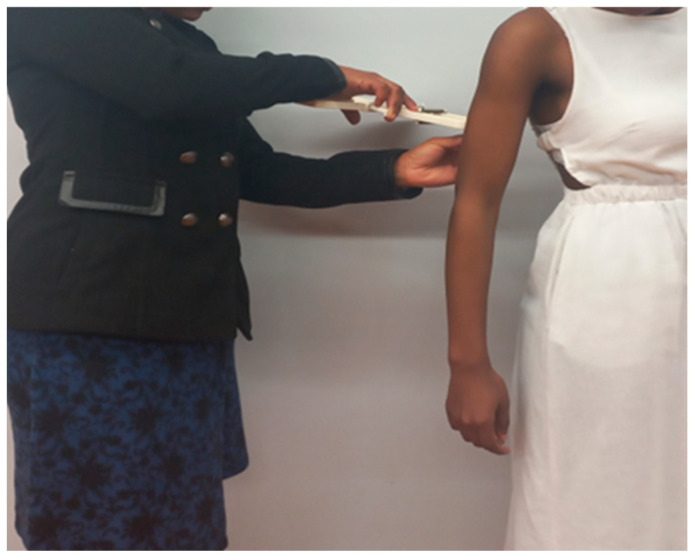
Triceps measurement.

**Figure 2 children-10-01113-f002:**
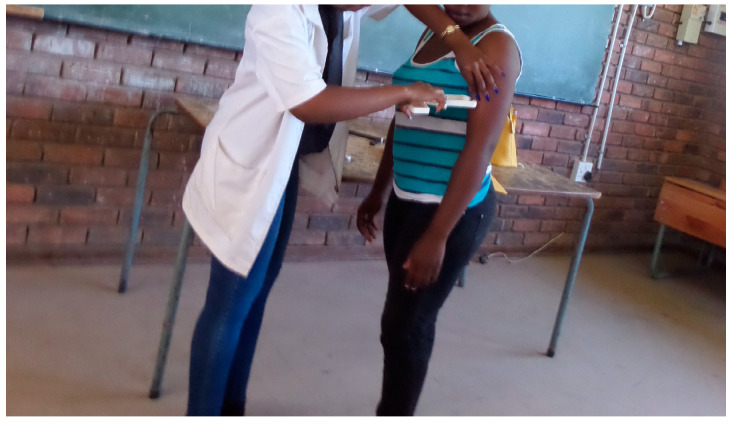
Biceps measurement.

**Figure 3 children-10-01113-f003:**
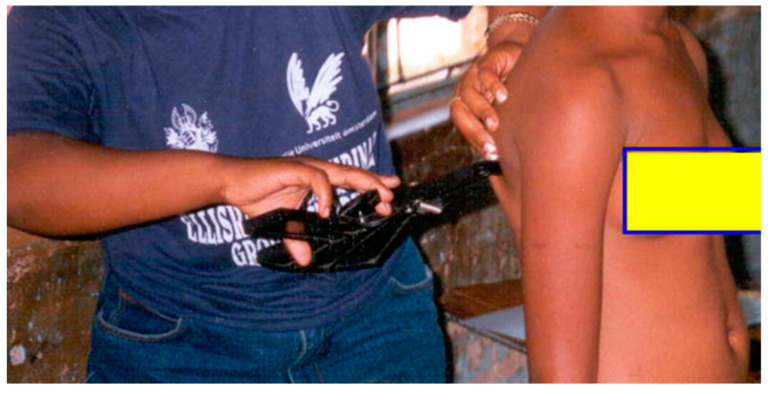
Subscapular measurement.

**Figure 4 children-10-01113-f004:**
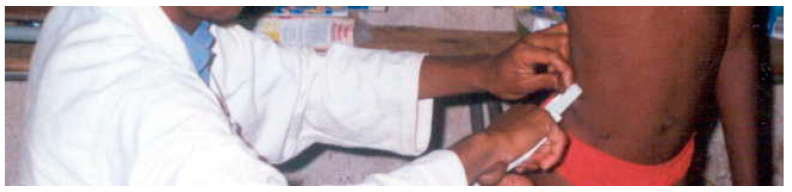
Suprailiac measurement.

**Table 1 children-10-01113-t001:** Descriptive statistics of Ellisras rural children and adolescents measured in autumn and spring (May 2000, November 2000).

Variables	N	Boys Autumn 2000	N	Boys Spring 2000	N	Girls Autumn 2000	N	Girls Spring 2000
	Median (25:75 IQR)							
Biceps								
5–10 years	413	4.0 ** (3.0:4.0)	335	4.0 ** (3.5:4.5)	378	5.5 * (4.5:6.5)	301	6.0 (4.5:7.0)
11–15 years	593	3.5 ** (3.0:4.0)	601	4.0 ** (3.5:4.5)	540	6.0 ** (5.0:7.0)	575	6.75 ** (5.5:8.0)
Subscapular								
5–10 years	413	5.0 (5.0:6.0)	335	5.0 (5.0:6.0)	378	6.0 ** (5.0:7.0)	301	6.5 ** (6.0:7.5)
11–15 years	593	5.5 (5.0:6.25)	599	6.0 (5.0:6.5)	540	7.0 ** (6.0:8.0)	575	8.0 ** (6.75:9.0)
Triceps								
5–10 years	413	6.5 (5.5:7.5)	335	6.5 (6.0:7.5)	378	7.88 (6.5:9.0)	301	8.0 (6.5:9.0)
11–15 years	593	6.75 ** (6.0:8.0)	599	7.0 ** (6.0:8.0)	540	9.0 * (7.5:11)	575	9.75 * (8.0:11.5)
Suprailiac								
5–10 years	413	4.0 (3.0:4.0)	335	4.0 (3.5:4.5)	378	5.0 (4.0:6.0)	301	5.0 (4.0:6.0)
11–15 years	593	4.0 ** (3.5:5.0)	599	4.25 ** (4.0:5.0)	540	6.0 ** (5.0:8.0)	575	6.75 ** (5.5:8.0)

** Shows significance level where *p* ≤ 0.03, * *p* ≤ 0.05.

**Table 2 children-10-01113-t002:** Prevalence of fat patterning of Ellisras rural children and adolescents measured in autumn and spring (May 2000, November 2000).

Variables	N	Boys Autumn 2000	N	Boys Spring 2000	N	Girls Autumn 2000	N	Girls Spring 2000
		n (%)		n (%)		n (%)		n (%)
Abdominal obesity by ST ratio								
5–10 years	413	33 (8)	335	24 (7)	378	17 (5) *	301	20 (7) *
11–15 years	593	63 (11) **	599	45 (8) **	540	16 (3) **	575	41 (7) **
Abdominal obesity by SST ratio								
5–10 years	413	22 (5)	335	22 (7)	378	19 (5)	301	16 (5)
11–15 years	593	25 (4) **	599	43 (7) **	540	26 (5)	575	33 (6)
Obesity by SSTB ratio								
5–10 years	413	23 (6)	335	19 (6)	378	13 (3) **	130	20 (7) **
11–15 years	593	29 (5)	599	33 (6)	540	30 (6)	575	26 (5)

* *p* ≤ 0.05, ** Shows significance level where *p* ≤ 0.03.

**Table 3 children-10-01113-t003:** Multinomial logistic regression for the association between obesity and seasons (autumn and spring 2000).

	**Autumn**			**Spring**		
Unadjusted						
	**Beta**	***p* Value**	**95% CI**	**Beta**	***p* Value**	**95% CI**
Abdominal obesity based on ST ratio	−2.633	<0.001 **	−2.822:−2.449	−2.559	<0.001 **	−2.738:−2.382
Abdominal obesity based on the SST ratio	−2.991	<0.001 **	−3.233:−2.803	−2.700	<0.001 **	−2.893:−2.518
Abdominal obesity based on the SSTB ratio	−2.958	<0.00 **	−3.191:−2.756	−2.860	<0.001 **	−3.087:−2.672
Adjusted for age and gender						
Abdominal obesity based on ST ratio	−1.084	0.054	−2.302:−0.075	−2.556	<0.001 **	−3.903:−1.432
Abdominal obesity based on the SST ratio	−2.897	<0.001 **	−4.331:−1.629	−2.718	<0.001 **	−4.165:−1.501
Abdominal obesity based on the SSTB ratio	−3.088	<0.001 **	−4.573:−1.828	−2.506	<0.001 **	−3.885:−1.346

** *p* < 0.01; CI = confidence interval.

## Data Availability

Data will be provided upon request.

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
