# Peer review of "Investigation of the Seasonal Variation in Fat Patterning of Ellisras Rural Children and Adolescents, in the Limpopo Province, South Africa"

_children, 2023, doi:10.3390/children10071113_

Round 1
Reviewer 1 Report
This is a very interesting study that has implications for those doing research in regions where seasonality is an issue.
In the abstract please define ST, SST, SSTB
In the Discussion:
In the discussion it is remarked (line 203) “One study showed that obesity prevalence increases with age 203 [20] which contradicts results from this study” There are many studies that show this and it is probably worth referencing.
Limitations
Skin fold was used and is totally acceptable, but has limitations in the severely obese
The main limitation of this study is that it only looks at one seasonal cycle. Since Spring 2000 preceded Fall 2000, it is not known whether time or seasonality or both played a role. This could be sorted out with multiple cycles. This should be mentioned as a limitation and called for in future research.
Reviewer 2 Report
Mualudzi et al investigated the seasonal variation in fat patterning of Ellis- 2 ras rural children and adolescents, in the Limpopo province, 3 South Africa. The manuscript was written nicely with sufficient data to support the conclusion.
I have few minor comments:
1. Additional figure to show the location of triceps, biceps, subscapular, and suprailiac would be advisable.
2. Regarding the influence of season on the physical activity, authors need to elaborate. First, about the climate in Ellisras or Lephalale. How many degrees difference between seasons or what is the temperature in each season. I raised this because if the temperature change was not too huge, the influence on the physical activity will not be too striking.
3. Authors reported that girls had significantly higher biceps, triceps, and supraspin the age 11-15. however, around 10-13 years of age, girls begin to grow taller and more rapidly than boys. Girls also reach puberty faster than boys. Could authors elaborate this point in and connect it with their findings because the observed results may be influenced by this point as well, not only by the season.
I hope the authors find my few minor comments useful to improve the manuscript.
Minor editing and checking is required.
Reviewer 3 Report
The manuscript is focused on the investigation of the seasonal variation in fat patterning of Ellisras area in South Africa, Limpopo province. The authors of the manuscript in this study used to determine it, they used skinfold measurements (suprailiac, subscapular, triceps, and biceps), there were obtained using standard procedures.
Introduction - I have no comments.
Material and methods - it is necessary to write what were the inclusion and exclusion criteria for the selection of samples - these data are missing.
It is necessary to add a more detailed description of the measurement of skin folds in the methodology.
- Measurements were taken in the year 2000 was used in this study (line 88). If the authors want to evaluate the current situation of childhood obesity, it is necessary to collect anthropometric data from the current period.
- it is necessary to specify the slim Guide skinfold calliper (manufacturer, country of origin)- line 93. - it would be methodologically appropriate to use nutritional protocols (to monitor the difference in energy intake and nutrients in the observed seasons), and it would also be valuable to use the detection of changes in children's physical activity in spring and autumn. Results - line 121 - incorrectly numbered chapter; the correct numbering should be 4.
- Table 1 – line 133 - formally, it is necessary to unify the size of the first letter in the name of the seasons; as well as text alignment in table cells. - similarly in Table 2 (line 146) and Table 3 (160)
Discussion - line 163 – incorrectly numbered chapter; the correct numbering should be 5.
- The discussion includes several citations of works in which blood parameters (triacylglycerols), adipose tissue hormones, monitoring the influence of nutrition and physical activity in the seasons - spring and autumn - were evaluated in relation to obesity. Were these parameters also monitored in the study described in this manuscript, if so, it would be appropriate to state them in the Results section. If not, for the correctness of the discussion, it would be appropriate to use a comparison only with the works of other authors who followed the measurement of skin folds. Conclusion - I have not comments.
References In the manuscript were used 29 literary sources (including 3 self-citations) of which only 6 sources are from the last 5 years; 9 literary sources are from the period of 5 to 10 years and 14 sources are older than 10 years. Therefore, it is necessary to update literary sources and add to more.
Minor editing of English language required
Reviewer 4 Report
The manuscript "Investigation of the seasonal variation in fat patterning of Ellisras rural children and adolescents, in the Limpopo province, South Africa" is still far from the scientifically soundness criteria.
1. On methods, data collection and collection is still unclear. Why is it data in 2000? and how the data was obtained in detail? What sampling technique method was used? Data sources must be clearly stated.
2. The justification for using the 2000 data considered for current publication must be logical. How does that reflect their current condition? while that time is far past. A comprehensive discussion regarding this justification should be provided.
3. Abstract is too long, must refer to the maximum limit determined by the Children's Journal (ISSN 2227-9067).
4. Extensive editing of English language required.
5. "fat patterning is associated with the seasons." In different seasons, of course, many factors influence, from diet and activity; why is there no data on diet, physical activity patterns and sleep quality? these three things are very important in their contribution to the nutritional status of children; let alone mental health.
This manuscript needs to be proofread by Native Speakers.
Round 2
Reviewer 4 Report
Figures obtained from publications or other sources (Not personal sources by Authors); therefore, Authors must provide evidence that they have been granted permission to use the figures in this publication.
English remains the same as before, there is no visible change (Extensive editing of English language required), and is not proven by the attachment of a language editing certificate.
Also, the comments below should not only be answered; but also discussed or added as justification in the main manuscript.
"Comment 1: On methods, data collection and collection is still unclear. Why is it data in 2000? and how the data was obtained in detail? What sampling technique method was used? Data sources must be clearly stated.
Response: This study used data from 2000 which is one of the oldest data from the Ellisras Longitudinal Study (ELS). Since the ELS follows the same participants overtime, the 2000 data will allow tracking of the ELS participants from childhood into adulthood since in 2000 the participants were between 5−15 years and currently, all ELS participants are in their adulthood. Furthermore, the 2000 data did not have a lot of missing values for all skinfolds measures."
English remains the same as before, there is no visible change (Extensive editing of English language required), and is not proven by the attachment of a language editing certificate.
